# Learning Universal Features for Generalizable Image Forgery Localization

## Abstract

In recent years, advanced image editing and generation methods have rapidly evolved, making detecting and locating forged image content increasingly challenging. Most existing image forgery detection methods rely on identifying the edited traces left in the image. However, because the traces of different forgeries are distinct, these methods can identify familiar forgeries included in the training data but struggle to handle unseen ones. In response, we present an approach for Generalizable Image Forgery Localization (GIFL). Once trained, our model can detect both seen and unseen forgeries, providing a more practical and efficient solution to counter false information in the era of generative AI. Our method focuses on learning general features from the pristine content rather than traces of specific forgeries, which are relatively consistent across different types of forgeries and therefore can be used as universal features to locate unseen forgeries. Additionally, as existing image forgery datasets are still dominated by traditional hand-crafted forgeries, we construct a new dataset consisting of images edited by various popular deep generative image editing methods to further encourage research in detecting images manipulated by deep generative models. Extensive experimental results show that the proposed approach outperforms state-of-the-art methods in the detection of unseen forgeries and also demonstrates competitive results for seen forgeries.

## 1 Introduction

Driven by the success of deep-generative models Goodfellow et al. (2020); Karras et al. (2019); Ho et al. (2020); Rombach et al. (2022), AI-based image manipulation tools have enabled realistic editing through simple interactions such as masks, sketches, and prompts Zeng et al. (2022); Rombach et al. (2022); Xie et al. (2023); Zeng et al. (2023); Epstein et al. (2023) that traditionally require sophisticated skills and tedious manual operations. Although they have undoubtedly brought numerous benefits, their widespread adoption has raised concerns about the credibility and trustworthiness of visual content. Consequently, forgery image detection and localization has emerged as an important research problem.

In recent years, a lot of effort has been made and a large number of methods have been proposed. Traditional methods are typically designed manually based on the observed artifacts of manipulated images such as noise patterns, JPEG artifacts, lens aberration, camera response function, color filter array Mahdian & Saic (2009); Amerini et al. (2011); Ferrara et al. (2012); Siwei et al. (2014); McCloskey & Albright (2018; 2019); Nikoukhah et al. (2019); Nataraj et al. (2019). Their applications are usually limited to the type of editing that produces the specific artifacts. In response, learning-based detection methods have been proposed to detect a wider range of forgeries by training a model on a large dataset of diverse forged images Wu et al. (2019); Dong et al. (2022); Liu et al. (2022).

Although learning-based approaches achieved excellent performance compared to traditional methods, their generalizability is still largely limited. They can accurately detect the types of forgeries included in training data, but often struggle to identify new forgeries to which they have not been exposed. In addition, the training datasets used in existing work are usually outdated and are still oriented towards basic manipulation techniques like splicing and copy-and-paste, despite that sophisticated editing methods powered by deep generative models have been widely applied nowadays. To detect image forgeries in practical applications, it is crucial to consider the use cases of

detecting forgeries produced by deep generative models and emphasize the generalization to unseen forgeries.

To this end, we investigate the generalizability of state-of-the-art forgery detection methods and present a comprehensive recipe for generalizable forgery detection in the deep learning era. First, we propose a new paradigm with a universal forgery detection network, which can generalize to unseen forgeries. We found that the inefficacy of existing methods in detecting unseen forgery is due to their heavy reliance on specific forgery traces of forged content. Since different editing methods may leave distinct traces, a model that looks for this trace can only recognize the type of forgery included in the training data.

Therefore, we encourage our detector to be more inclined towards learning and utilizing authentic image features in the pristine areas instead, which are more consistent across different types of forgeries and can be used as universal features to learn a generalizable detector. More specifically, our method supervises the encoder feature using features from masked images in which the forged areas are removed. The decoder then utilizes these features in conjunction with other general features to produce the localization mask.

Second, to encourage future research on the detection of advanced image forgeries, we create a new forged image dataset, Forgery ADE. This dataset consists of a total of 177,680 images partially manipulated using eight popular state-of-the-art image editing and generation methods based on GANs and diffusion models. For each original image, we create different variants of forged images applying different types of forgeries in the same area. This makes it easier to study the effect of different types of training forgery on test-time detection robustness and compare detection performance between different types of forgery.

Furthermore, we study and addressed several pervasive data-related issues such as the influence of semantic correlation between forgery and image content, false positives on authentic images, the influence of the number of training forgery types on the model's generalization ability, and the impact of masking, a common post-processing step for deep learning based image editing methods, on forgery detection performance and generalization.

We summarize our contributions as follows:

- We proposed a generalizable image forgery detection and localization method capable of detecting unseen forgeries, while also showing competitive performance compared to existing methods in detecting seen forgeries.

- We construct a new image forgery dataset using various advanced image manipulation and generation methods, which is more suitable for universal and cross-method forgery detection research than most existing datasets.

- We addressed several data-related issues in existing image forgery detection and localization methods by introducing a comprehensive training data configuration.

## 2 RELATED WORK

Many methods have been proposed to assess the authenticity of images McCloskey & Albright (2018); Marra et al. (2018); Rossler et al. (2019); Nataraj et al. (2019); McCloskey & Albright (2019); Li et al. (2020). However, these approaches frequently exhibit unsatisfactory performance when applied to images forged by unseen editing methods. Some attempts have been made to address this challenge. Wang et al. (2020) demonstrated that GAN-based generated images share some common systematic flaws, which can be distinguished from real images by classifiers trained on a specific generator. Zeng et al. (2017); Zhang et al. (2019); Frank et al. (2020) found that the GAN-based image generation method is more likely to expose common artifacts in the spectral domain, which can be used to distinguish them from authentic images. Wang et al. (2023) tries to identify the images generated by the diffusion model Ho et al. (2020) based on reconstruction errors of a pre-trained diffusion model. However, these methods can only determine the authenticity of the entire image and cannot locate the forged areas in the image.

Some work focuses on locating specific forged regions in images by modeling it as a pixel-level binary classification problem. Liu et al. (2022); Dong et al. (2022) utilize the learned multi-scale

Figure 1: Illustration of the traditional classification-based image forgery detection pipeline (left) and our proposed GIFL method (right).

and multi-view to detect forged areas. Kwon et al. (2022) designed CAT-Net to locate slicing and copy-and-paste forgeries from the perspective of detecting the anomaly of JPEG compression artifacts. Some research is dedicated to finding inpainting forgery in images Zhu et al. (2018); Li & Huang (2019); Lu & Niu (2020); Kumar & Meenpal (2021); Zhu et al. (2023); Zhang et al. (2022); Wu & Zhou (2021).

Some studies have focused on detecting both general and composite image forgeries. Wu et al. (2019) proposed ManTra-Net, which uses a long short-term memory solution to locate various types of forgery traces. Zhuang et al. (2021) proposed a fully convolutional network to detect the commonly used editing operations in Photoshop, and designed a training data generation strategy based on Photoshop scripts. Wu et al. (2022) proposed a method based on noise-modeling and a robust training scheme for detecting forged images that are shared on online social media. Guillaro et al. (2023) proposed TruFor, which utilizes learned noise-sensitive fingerprints to detect manipulation traces. Wu et al. (2023) proposed FOCAL, an image forgery detection method based on pixel-level contrastive learning and unsupervised clustering. Zhai et al. (2023) proposed WSCL, which aims to enhance generalization ability through weakly-supervised learning. Recently, Guo et al. (2024) proposed EITLNet, which effectively locates various image forgeries through the enhanced two-branch transformer encoder with attention-based feature fusion.

Although these methods have broadened the scope of forgery detection, they still rely heavily on locating the specific traces of the forgeries in training data and are prone to severe degradation in detecting unseen forgeries.

## 3 METHOD

### 3.1 LEARNING UNIVERSAL FEATURES FOR GENERALIZABLE FORGERY LOCALIZATION

A forged image consists of two parts: the authentic part and the forged part. Given a forged image $I$, existing approaches use a classifier $C$ to perform pixel-level binary classification. The goal of the classifier is to assign different labels, *e.g.* $0, 1$, to pixels in the two parts, and produce a binary mask $M_d$:

$$M_d = C(I).$$

Through learning on a large dataset of images altered by various editing methods, a powerful deep neural network classifier can learn to recognize manipulation traces and detect many types of forgeries. However, these classifiers often rely heavily on the specific traces left by forgery methods in the forged content, making it challenging to detect novel forgeries that have not been encountered. This is a critical issue because image editing methods evolve rapidly and new techniques are constantly being developed.

In this work, we aim to encourage the detector to pay more attention to the general information shared among different types of forgeries rather than specific forgery traces to develop a more generalizable approach. We observe that while forged parts may have distinct traces due to various editing methods, authentic regions remain relatively consistent. Therefore, we propose to reconstruct the features of the authentic area that are devoid of forged content to make the model $R$ more inclined to learn and utilize the authentic information:

$$F_r = R(F_I)$$

where $F_r$ denotes the reconstructed authentic feature and $F_I$ is the entire image feature encoded by a frozen encoder.

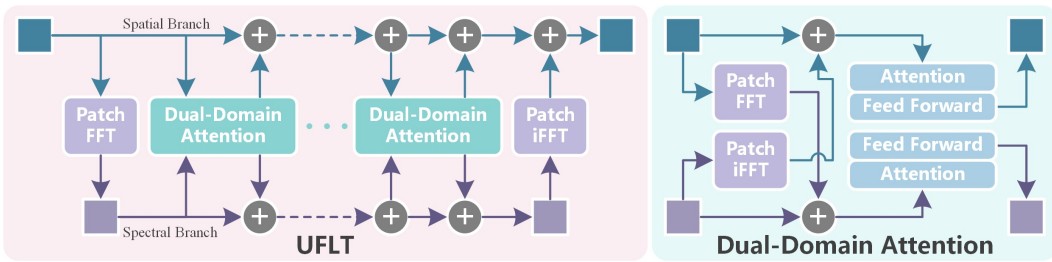

Figure 2: Architecture of our proposed Universal Forgery Localization Transformer (UFLT) (red part), which contains multiple dual-domain attention components (cyan part).

Fig. 1 illustrates the overall pipeline of most existing approaches and the proposed method. In essence, we transform the forgery localization task from a binary classification task that distinguishes between real and forged content to a regression task centered on authentic feature reconstruction. Using the wider metric space of a regression process, more useful information can be retained and used to achieve better performance and robustness.

Moreover, we decouple the detection process from the encoding and decoding process inspired by Chen et al. (2024). We utilize the same frozen encoder $E$ to encode the features of both the input image and the partial images where the forgery areas are removed:

$$F_I = E(I)$$

$$F_t = E(I \odot (1 - M_t))$$

where $\odot$ represents element-wise multiplication. We then utilize a Universal Forgery Localization Transformer (UFLT), which will be detailed in Sec. 3.2, to align the features of the forged image with those of the partial image. This alignment reduces the gap between encoded features of different forgeries, leading to improved generalization ability.

Finally, a fully connected layer is used as a decoder to obtain the final detection mask based on the reconstructed features:

$$M_d = FC(F_r)$$

The loss function of our method comprises two parts: utilizing L2 loss to guide the regressor's feature reconstruction and alignment with input features, and supervising the final mask output through a combination of BCE loss and IoU loss Zhou et al. (2019):

$$\mathcal{L} = \mathcal{L}_2(F_r, F_t) \times 10 + \mathcal{L}_{BCE}(M_d, M_t) + \mathcal{L}_{IoU}(M_d, M_t)$$

## 3.2 UNIVERSAL FORGERY LOCALIZATION TRANSFORMER

It has been pointed out that the spectral feature traces of different types of forgery may appear in different depth of a network Zeng et al. (2017); Frank et al. (2020); Zhang et al. (2019); Xu et al. (2019). However, existing spectral detection networks Kwon et al. (2022); Wang et al. (2022a); Zhou et al. (2024) typically use a manually designed feature extractor to obtain traces from a specific layer through a single spectral transform and process the spectral features as another branch independent of the spatial domain pipeline, which means that these methods can only extract the spectral traces of some specific forgeries. Moreover, their single spectral transform and independent branch processing result in the isolation of spectral information during intermediate processing, with no interaction with spatial information before the inverse transform.

To this end, we propose a Universal Forgery Localization Transformer (UFLT), as depicted in Fig. 2. To effectively exploit the traces exposed at different depths in different domains, we draw inspiration from FFC Chi et al. (2020) and establish the interaction and fusion of features across domain and across depth by connecting the paths of the two domains in each layer of the network. The transformer encoder and feedforward network architectures are identical to ViT Dosovitskiy et al. (2020).

In contrast to FFC, which utilizes an image-level Fourier transform to capture global information, our approach focuses on leveraging local spectral domain information along with spatial positional relationships to extract local information. To ensure compatibility with the vision transformer network and make full use of its global self-attention capabilities for establishing long-distance correlations, UFLT performs the Fourier transform on the patch embedding scale. Specifically, letting $X_{pix} \in \mathbb{R}^{N \times D}$ be the input feature, where $N$ and $D$ are the number and dimension of embeddings, we apply a 2-D fast Fourier transform (FFT) to embeddings, and then concatenate the imaginary and real parts of the spectral features to produce $X_{freq} \in \mathbb{R}^{N \times (D \times 2)}$. Due to the redundancy in the conjugate symmetric signal obtained from the 2-D FFT, we can reduce the dimension of $X_{freq}$ without losing essential information. Therefore, we transform $X_{freq}$ into a $D$ dimensional feature through a fully connected layer to ensure that its shape aligns with that of $X_{pix}$, allowing the vision transformer to use features of the spectral domain.

## 4 FORGERY ADE DATASET

Diversifying the type of forgeries in training is an effective way to enhance the generalizability of forgery detectors Wang et al. (2020). However, most of the existing forgery image detection and localization datasets Kwon et al. (2022); Dong et al. (2013); De Carvalho et al. (2013); Ng et al. (2004); Wen et al. (2016) lack diversity and are still dominated by images produced using manual slicing. These outdated and diversity limited training datasets are not suitable for contemporary deep learning based image manipulation methods.

Although some recently developed datasets have included deep learning base forgeries, they still have several critical issues. First, in most datasets Wen et al. (2016); Guillaro et al. (2023); Jia et al. (2023); Sun et al. (2023), the location, shape, and contents of the forged areas are determined based on the semantics of the image. However, in practical applications, forgery may appear in any shape in any region of the image and may be unrelated to semantics. Therefore, training on these forgery images with semantic connections can lead detectors to rely on semantics to detect the forgery, resulting in reduced performance on forgeries without semantic connections. Second, most existing datasets only include forged images and lack authentic images as negative samples. As a result, false positives often occur when the input is a clean image without any forgery VidalMata et al. (2023). Moreover, we found that many existing models learned a shortcut to detecting deep learning based forgeries by detecting the seam between the manipulated area and the pristine area. This is because most deep-generative inpainting models have a post-processing step that blends the model output with the original image using the inpainting mask. We present a detailed study of these issues in Section 5.

To this end, we create a new forged image dataset based on the ADE 20K Zhou et al. (2017) dataset, called Forgery ADE. Examples of forged images generated by different forgery methods in the dataset are shown in Fig. 4.

We select eight of the most popular and representative deep learning based image inpainting methods as forgery approaches, including 4 GAN-based methods: Deepfill v2 Yu et al. (2019), CTSDG Guo et al. (2021), CR-Fill Zeng et al. (2021), LaMa Suvorov et al. (2022), and 4 diffusion-based methods: LDM Rombach et al. (2022), SSDE Song et al. (2020), DDNM Wang et al. (2022b), RePaint Lugmayr et al. (2022). Each of these methods is applied to all 20,210 training images and 2,000 test images in the ADE 20K, producing eight sets of forgery images for a total of 177,680 images. We carefully selected a set of irregular occlusion masks provided by Zeng et al. (2021) that are not related to the image and randomly rotated and flipped them to increase the diversity of the data and further avoid semantic association. We scale all images and masks to $512 \times 512$. All forged images are the direct output of the generated model without post-processing masking. In training, we provide authentic images as negative samples, of which the ground truth is all zero maps.

## 5 EXPERIMENTS

### 5.1 IMPLEMENTATION DETAILS

Considering the availability of code and the popularity of forgery methods, we use the most popular GAN-based forgery method Deepfill v2 Yu et al. (2019) and the diffusion-based forgery method

Table 1: Quantitative comparison. Bold for best results.

| Prior | Forgery | Metric | ManTra-Net | MVSS-Net | PSCC-Net | CAT-Net | IID-Net | IF-OSN | IML-ViT | TruFor | FOCAL | EITLNet | GIFL |
|---|---|---|---|---|---|---|---|---|---|---|---|---|---|
| Trained | Deepfill v2 | F1 | 0.8855 | **0.9117** | 0.8765 | 0.8532 | 0.8583 | 0.9069 | 0.9035 | 0.8779 | 0.8239 | 0.8956 | 0.8539 |
| | | IOU | 0.7071 | **0.7719** | 0.7009 | 0.6523 | 0.6633 | 0.7656 | 0.7166 | 0.7057 | 0.6086 | 0.7362 | 0.6446 |
| | | ACC | 0.9805 | 0.9881 | 0.9815 | 0.9665 | 0.9795 | **0.9888** | 0.9865 | 0.9852 | 0.9519 | 0.9847 | 0.9756 |
| | | AUC | 0.8905 | **0.9133** | 0.8718 | 0.8724 | 0.8565 | 0.9010 | 0.9082 | 0.8611 | 0.8251 | 0.9030 | 0.8647 |
| | LDM | F1 | 0.5623 | 0.7063 | 0.6382 | 0.5019 | 0.5951 | 0.6480 | 0.5193 | 0.6069 | 0.5238 | **0.7218** | 0.7078 |
| | | IOU | 0.1262 | 0.3703 | 0.3108 | 0.0547 | 0.1979 | 0.2768 | 0.0795 | 0.2106 | 0.0821 | **0.4015** | 0.3655 |
| | | ACC | 0.8842 | **0.9283** | 0.8228 | 0.8741 | 0.8936 | 0.9074 | 0.8613 | 0.9124 | 0.8546 | 0.9183 | 0.9248 |
| | | AUC | 0.5672 | 0.7193 | 0.7147 | 0.5273 | 0.6171 | 0.6576 | 0.5511 | 0.6103 | 0.5409 | **0.7527** | 0.7278 |
| | CASIA1.0 | F1 | 0.4799 | 0.6526 | 0.5431 | 0.4902 | 0.4848 | 0.6723 | 0.5259 | 0.6117 | 0.5628 | 0.5711 | **0.7827** |
| | | IOU | 0.0047 | 0.3060 | 0.1066 | 0.0192 | 0.0147 | 0.3482 | 0.0767 | 0.2183 | 0.1415 | 0.1490 | **0.5010** |
| | | ACC | 0.9114 | 0.9272 | 0.8917 | 0.9087 | 0.9072 | 0.9383 | 0.9093 | 0.9372 | 0.8807 | 0.9224 | **0.9490** |
| | | AUC | 0.5015 | 0.6636 | 0.5677 | 0.5083 | 0.5057 | 0.6795 | 0.5430 | 0.6131 | 0.5701 | 0.5821 | **0.7934** |
| | Seen AVG | F1 | 0.6426 | 0.7569 | 0.6859 | 0.6151 | 0.6461 | 0.7424 | 0.6496 | 0.6988 | 0.6368 | 0.7295 | **0.7815** |
| | | IOU | 0.2793 | 0.4827 | 0.3728 | 0.2421 | 0.2920 | 0.4635 | 0.2909 | 0.3782 | 0.2774 | 0.4289 | **0.5037** |
| | | ACC | 0.9254 | 0.9479 | 0.8987 | 0.9164 | 0.9268 | 0.9448 | 0.9190 | 0.9449 | 0.8957 | 0.9418 | **0.9498** |
| | | AUC | 0.6531 | 0.7654 | 0.7181 | 0.6360 | 0.6598 | 0.7460 | 0.6674 | 0.6948 | 0.6454 | 0.7459 | **0.7953** |
| Unseen | CTSDG | F1 | 0.5619 | 0.6831 | 0.5650 | 0.5241 | 0.5667 | 0.6412 | 0.5421 | 0.5212 | 0.5258 | 0.5513 | **0.7963** |
| | | IOU | 0.1269 | 0.3321 | 0.1878 | 0.0825 | 0.1547 | 0.2632 | 0.1106 | 0.0800 | 0.0890 | 0.1313 | **0.5277** |
| | | ACC | 0.8667 | 0.9247 | 0.8032 | 0.8749 | 0.8695 | 0.9082 | 0.8692 | 0.8834 | 0.8512 | 0.8861 | **0.9545** |
| | | AUC | 0.5743 | 0.6839 | 0.6173 | 0.5449 | 0.5862 | 0.6429 | 0.5700 | 0.5408 | 0.5416 | 0.5722 | **0.8209** |
| | CR-Fill | F1 | 0.5481 | 0.6808 | 0.6041 | 0.5487 | 0.5858 | 0.5902 | 0.5326 | 0.5640 | 0.5509 | 0.6231 | **0.7876** |
| | | IOU | 0.1031 | 0.3261 | 0.2096 | 0.1189 | 0.1802 | 0.1824 | 0.0960 | 0.1527 | 0.1192 | 0.2511 | **0.5086** |
| | | ACC | 0.8874 | 0.9317 | 0.8992 | 0.8840 | 0.8980 | 0.9036 | 0.8721 | 0.8988 | 0.8649 | 0.9096 | **0.9525** |
| | | AUC | 0.5536 | 0.6760 | 0.6175 | 0.5662 | 0.6015 | 0.5958 | 0.5589 | 0.5780 | 0.5607 | 0.6389 | **0.8070** |
| | LaMa | F1 | 0.4735 | 0.5137 | 0.4734 | 0.5073 | 0.5481 | 0.4788 | 0.5152 | 0.5127 | 0.5197 | 0.5591 | **0.6580** |
| | | IOU | 0.0125 | 0.0732 | 0.0152 | 0.0638 | 0.1230 | 0.0238 | 0.0728 | 0.0742 | 0.0818 | 0.1557 | **0.2884** |
| | | ACC | 0.8717 | 0.8935 | 0.8733 | 0.8761 | 0.8931 | 0.8789 | 0.8681 | 0.8868 | 0.8482 | 0.8987 | **0.9279** |
| | | AUC | 0.5059 | 0.5370 | 0.5084 | 0.5316 | 0.5668 | 0.5119 | 0.5445 | 0.5373 | 0.5366 | 0.5803 | **0.6596** |
| | SSDE | F1 | 0.4781 | 0.5082 | 0.5376 | 0.4704 | 0.4883 | 0.4879 | 0.5020 | 0.4720 | 0.5028 | 0.4903 | **0.6779** |
| | | IOU | 0.0192 | 0.0609 | 0.1193 | 0.0108 | 0.0381 | 0.0348 | 0.0565 | 0.0126 | 0.0586 | 0.0380 | **0.3216** |
| | | ACC | 0.8688 | 0.8840 | 0.8606 | 0.8655 | 0.8663 | 0.8785 | 0.8618 | 0.8744 | 0.8465 | 0.8778 | **0.9188** |
| | | AUC | 0.5176 | 0.5311 | 0.5725 | 0.5026 | 0.5235 | 0.5176 | 0.5336 | 0.5063 | 0.5183 | 0.5217 | **0.7010** |
| | DDNM | F1 | 0.5258 | 0.5511 | 0.5668 | 0.4861 | 0.5656 | 0.5252 | 0.5508 | 0.4901 | 0.5467 | 0.5276 | **0.6589** |
| | | IOU | 0.0741 | 0.1266 | 0.1575 | 0.0300 | 0.1503 | 0.0889 | 0.1231 | 0.0355 | 0.1118 | 0.0929 | **0.2949** |
| | | ACC | 0.8861 | 0.9018 | 0.8813 | 0.8706 | 0.8934 | 0.8916 | 0.8789 | 0.8812 | 0.8640 | 0.8909 | **0.9263** |
| | | AUC | 0.5369 | 0.5676 | 0.5945 | 0.5130 | 0.5875 | 0.5474 | 0.5732 | 0.5179 | 0.5542 | 0.5483 | **0.6698** |
| | RePaint | F1 | 0.4923 | 0.5335 | 0.5892 | 0.4902 | 0.5484 | 0.5380 | 0.4986 | 0.5166 | 0.5102 | 0.5521 | **0.6478** |
| | | IOU | 0.0354 | 0.1001 | 0.1924 | 0.0383 | 0.1252 | 0.1095 | 0.0505 | 0.0778 | 0.0674 | 0.1384 | **0.2721** |
| | | ACC | 0.8691 | 0.8890 | 0.8835 | 0.8706 | 0.8932 | 0.8932 | 0.8615 | 0.8873 | 0.8482 | 0.8947 | **0.9172** |
| | | AUC | 0.5169 | 0.5519 | 0.6112 | 0.5178 | 0.5725 | 0.5566 | 0.5319 | 0.5392 | 0.5285 | 0.5735 | **0.6594** |
| | COVERAGE | F1 | 0.4764 | 0.4830 | 0.4706 | 0.4755 | 0.4700 | 0.4838 | 0.4848 | 0.4730 | 0.4902 | 0.4737 | **0.5647** |
| | | IOU | 0.0097 | 0.0169 | 0.0041 | 0.0089 | 0.0027 | 0.0236 | 0.0275 | 0.0045 | 0.0453 | 0.0052 | **0.1417** |
| | | ACC | 0.8797 | 0.8857 | 0.8787 | 0.8797 | 0.8806 | 0.8859 | 0.8639 | 0.8884 | 0.8422 | 0.8851 | **0.8971** |
| | | AUC | 0.5008 | 0.5080 | 0.4982 | 0.5002 | 0.4987 | 0.5111 | 0.5086 | 0.5036 | 0.5085 | 0.5007 | **0.5763** |
| | CocoGlide | F1 | 0.4253 | 0.4673 | 0.4183 | 0.4249 | 0.4186 | 0.4224 | 0.4509 | 0.4365 | 0.4538 | 0.4789 | **0.5712** |
| | | IOU | 0.0122 | 0.0784 | 0.0069 | 0.0120 | 0.0047 | 0.0104 | 0.0506 | 0.0305 | 0.0511 | 0.0805 | **0.2148** |
| | | ACC | 0.7460 | 0.7413 | 0.7429 | 0.7446 | 0.7494 | 0.7520 | 0.7415 | 0.7466 | 0.7363 | 0.7626 | **0.7968** |
| | | AUC | 0.5021 | 0.5312 | 0.5011 | 0.5028 | 0.5023 | 0.5054 | 0.5220 | 0.5122 | 0.5161 | 0.5383 | **0.6114** |
| | Unseen AVG | F1 | 0.4977 | 0.5526 | 0.5281 | 0.4909 | 0.5239 | 0.5209 | 0.5096 | 0.4983 | 0.5125 | 0.5320 | **0.6703** |
| | | IOU | 0.0491 | 0.1393 | 0.1116 | 0.0457 | 0.0974 | 0.0921 | 0.0735 | 0.0585 | 0.0780 | 0.1116 | **0.3212** |
| | | ACC | 0.8594 | 0.8815 | 0.8528 | 0.8583 | 0.8668 | 0.8740 | 0.8521 | 0.8684 | 0.8377 | 0.8757 | **0.9114** |
| | | AUC | 0.5247 | 0.5733 | 0.5651 | 0.5224 | 0.5549 | 0.5486 | 0.5428 | 0.5294 | 0.5335 | 0.5588 | **0.6882** |

LDM Rombach et al. (2022) to create edited images in Forgery ADE. We also include a representative traditional splicing forgery image dataset CASIA v2 Dong et al. (2013) as training data. In addition, we add unmanipulated clean images to the training data as negative samples, with a ratio of 1 : 1 to forged images.

We use the pre-trained DINOv2 ViT-L/14 Oquab et al. (2023) as the encoder, which is frozen during training, and use a fully connected layer as decoder. UFLT adopts the same configuration as ViT-L/14 Dosovitskiy et al. (2020). UFLT and the decoder are optimized together by the Adam optimizer. Training is performed on images with a resolution of $448 \times 448$, with all data enhancement measures in Wang et al. (2020) and Wu et al. (2023). The learning rate is set to $1e-4$ and the batch size is 8. We use the Pytorch framework for our implementation and train on a Nvidia A100 GPU.

## 5.2 PERFORMANCE OF OUR APPROACH

We compared our proposed approach with several state-of-the-art image forgery detection and localization methods, including EITLNet Guo et al. (2024), FOCAL Wu et al. (2023), TruFor Guillaro et al. (2023), IML-ViT Ma et al. (2023), IF-OSN Wu et al. (2022), IID-Net Wu & Zhou (2021), CAT-Net Kwon et al. (2022), PSCC-Net Liu et al. (2022), MVSS-Net Dong et al. (2022) and ManTra-Net Wu et al. (2019). To ensure fair comparison, all methods are implemented in accordance with the original paper, while trained on the same dataset with the same data augmentation measures as ours.

A comparative experiment is conducted on all eight forgery test sets in Forgery ADE, as well as two splicing forgery datasets CASIA v1 Dong et al. (2013) and COVERAGE Wen et al. (2016), and an object synthetic dataset CocoGlide Guillaro et al. (2023). Performance is evaluated using pixel-level metrics including F1, IoU, ACC, and AUC.

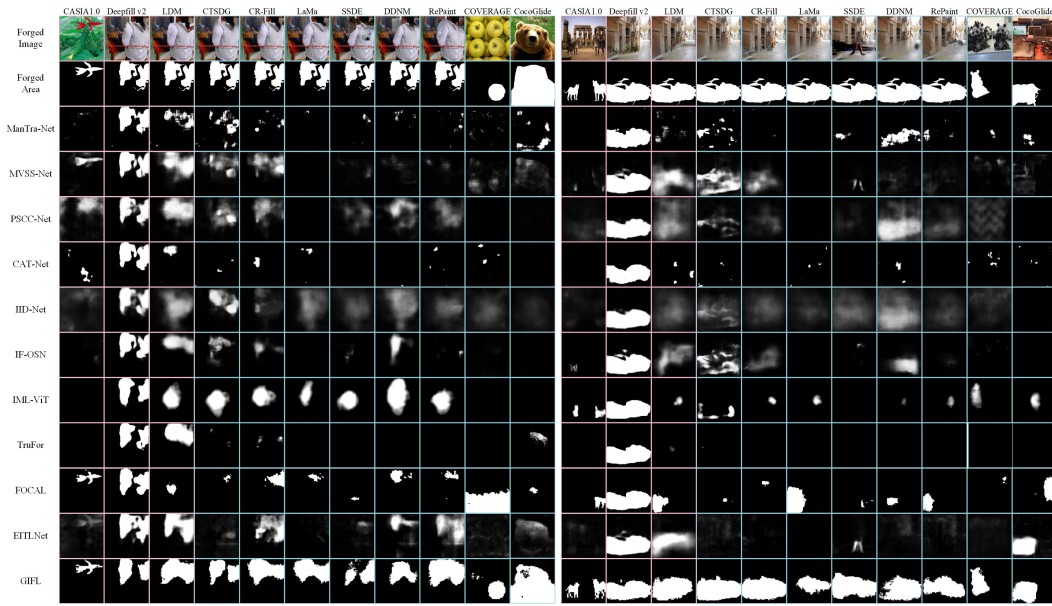

Figure 3: Visual comparison of our results and those of previous methods. Red boxes indicate the results on seen forgeries and blue for unseen ones. Zoom-in to see the details. More results are included in the Appendix.

As illustrated in Table 1, GIFL achieves state-of-the-art performance in detecting unseen forgeries and also shows competitive performance on seen forgeries. In particular, GIFL demonstrates significantly superior performance over other methods in detecting highly realistic images edited by diffusion-based methods.

Fig. 3 shows the detection results of our method and others. More visual comparisons are shown in the Appendix (Fig. 5). It can be observed that existing methods generally fail to achieve satisfactory localization results for unseen forgeries, many forged contents cannot be accurately and completely detected, and authentic contents are often incorrectly labeled as forgeries. Some methods tend to focus on a specific one in multiple seen forgeries and fail to detect others. In contrast, GIFL provides more accurate results in unseen forgeries and performs more consistently and reliably on multiple trained forgeries.

## 5.3 ABLATION STUDY

We investigate the specific effects of each component of our approach. The following experiments adopt the same model parameters and experimental settings as stated in Sec. 5.1. To speed up the experiment, we set the encoder and UFLT to the configuration of ViT-B, trained on forged images of Deepfill v2 and LDM with the same number of authentic images, and tested only on the first 50 images of each forgery in Forgery ADE. The results are shown in Table 2. A more complete analysis can be found in the Appendix (Table 6).

**Learning Method.** We design a series of experiments to explore the effectiveness of our proposed image forgery localization method. Firstly, we train the UFLT and decoder using the traditional mask-targeted classification training method as a comparison (Option I). Then, we change the target for the reconstruction in the GIFL to the feature of the target mask encoded by the encoder:

$$F_t = E(M_t)$$

thus independently applying the feature space alignment strategy without the reconstruction of the image features to show the influence of each component (Option II). By comparing the results of Option II with those of Option I and the baseline, we can see that the authentic feature-reconstruction approach and the feature space alignment strategy each brings a significant improvement on unseen but also cause a slight performance degradation on seen forgeries.

Table 2: Ablation study on different configurations of GIFL.

| Category | Option | Seen Average | | | | Unseen Average | | | |
|---|---|---|---|---|---|---|---|---|---|
| | | F1 | IOU | ACC | AUC | F1 | IOU | ACC | AUC |
| Baseline | | 0.8048 | 0.5566 | 0.9321 | 0.8253 | 0.7226 | 0.4059 | 0.9016 | 0.7362 |
| Method | I | 0.8212 | 0.5849 | 0.9346 | 0.8287 | 0.6653 | 0.3085 | 0.8845 | 0.6704 |
| | II | 0.8170 | 0.5772 | 0.9340 | 0.8258 | 0.6748 | 0.3243 | 0.8805 | 0.6834 |
| | III | 0.7485 | 0.4565 | 0.9171 | 0.7621 | 0.6657 | 0.3151 | 0.8884 | 0.6761 |
| | IV | 0.8049 | 0.5585 | 0.9346 | 0.8152 | 0.5681 | 0.1650 | 0.8622 | 0.5847 |
| Network | V | 0.7872 | 0.5241 | 0.9302 | 0.7909 | 0.6752 | 0.3377 | 0.8921 | 0.6843 |
| | VI | 0.7625 | 0.4905 | 0.8801 | 0.8130 | 0.7206 | 0.4152 | 0.8641 | 0.7658 |
| | VII | 0.7655 | 0.4819 | 0.9253 | 0.7706 | 0.6320 | 0.2613 | 0.8772 | 0.6405 |
| | VIII | 0.7724 | 0.5011 | 0.9203 | 0.7921 | 0.7030 | 0.3794 | 0.8959 | 0.7201 |
| | IX | 0.7918 | 0.5387 | 0.9090 | 0.8344 | 0.7236 | 0.4164 | 0.8763 | 0.7599 |

To investigate the impact of specifically reconstructing authentic features in images, we reconstruct the feature of the complete image (Option III):

$$F_t = E(I)$$

Further, we replace the targeted features with the forgery feature of the image:

$$F_t = E(I \odot M_t)$$

so that UFLT learns to reconstruct the content of the forged area rather than the authentic one (Option IV). Option III proves that reconstructing authentic features plays a crucial role. The performance of Option IV in unseen forgeries is far below the baseline, which shows the clear advantage of using authentic content for generalizable forgery detection.

**Universal Forgery Localization Transformer.** To verify the effectiveness of UFLT, we first construct an UFLT-spatial that is implemented only in the spatial domain, which has the same structure and number of parameters as the baseline, but without any spectral transformation (Option V). Then we performed the spectral transformation on the input and output of the network, thereby constructing an UFLT-spectral that is only in the spectral domain (Option VI). According to the results, although UFLT performed solely in the spectral domain has certain advantages, its performance gain is still limited. This shows that the benefit of UFLT comes largely from the fusion and utilization of information from both domains.

**Patch-Level Domain Transformation.** To verify the effectiveness of patch-level domain transformation, we introduced three spectral transformation schemes for comparison: 2-D FFT on the entire feature at the image level (Option VII), $32 \times 32$ (Option VIII) and $8 \times 8$ (Option IX) scales, respectively. Since the spectral domain features are obtained by transformation at the image level, which does not retain any spatial position information, it hardly brought any performance improvement for forgery localization tasks. Although the transformation scheme on the $32 \times 32$ scale preserves information in local regions, its performance is still inferior to the baseline due to its inability to effectively utilize the global attention mechanism of ViT. The $8 \times 8$ scheme yields a similar performance to the baseline but has higher computational complexity.

### 5.4 DATA-RELATED STUDIES

In this section, we study the impact of several data-related issues on detection performance and generalization. The experimental settings and model configuration in this section are the same as those in the ablation studies, except for the training data. We measure the false positive errors on authentic images using pixel-level ACC (p-ACC) and image-level ACC (i-ACC). The experimental results are shown in Table 3 and Table 4.

Additionally, we investigate the impact of various image quality degradation (such as compression, resizing, *etc.*) on detection performance. Please refer to Appendix A.2 for details.

**Semantic Correlations.** To investigate the impact of semantic correlation between forgery and image content on detection performance, we use images generated by LaMa Suvorov et al. (2022) from GRE Sun et al. (2023) and the LaMa subset of Forgery ADE for training, which were semantically related and nonrelated, respectively. We then test them on two datasets with semantic correlation, Guillaro et al. (2023); Jia et al. (2023); Sun et al. (2023) and Wen et al. (2016), and two datasets

Table 3: Performance comparison under different dataset settings, bold for trained forgeries.

| Forgery | Metric | Baseline | Data Diversity | | | | | False Positives | | | Masking | | |
|---|---|---|---|---|---|---|---|---|---|---|---|---|---|
| | | | I | II | III | IV | V | VI | VII | VIII | IX | X | XI |
| Deepfill v2 | F1 | **0.8932** | **0.9051** | 0.7320 | **0.8874** | **0.8859** | **0.8810** | **0.8968** | **0.9035** | **0.8982** | 0.8641 | 0.8722 | **0.9029** |
| | IOU | **0.7195** | **0.7433** | 0.4243 | **0.7075** | **0.7025** | **0.6909** | **0.7285** | **0.7419** | **0.7271** | 0.6647 | 0.6758 | **0.7415** |
| | ACC | **0.9650** | **0.9695** | 0.9007 | **0.9617** | **0.9620** | **0.9608** | **0.9678** | **0.9698** | **0.9677** | 0.9595 | 0.9510 | **0.9688** |
| | AUC | **0.9093** | **0.9161** | 0.7564 | **0.9055** | **0.9069** | **0.8979** | **0.9117** | **0.9122** | **0.9065** | 0.8576 | 0.8985 | **0.9124** |
| CTSDG | F1 | 0.8032 | 0.7500 | 0.7216 | **0.8731** | **0.8629** | **0.8621** | 0.7940 | 0.8079 | 0.7864 | 0.8061 | 0.8303 | 0.8720 |
| | IOU | 0.5464 | 0.4412 | 0.4062 | **0.6831** | **0.6654** | **0.6649** | 0.5252 | 0.5537 | 0.5206 | 0.5539 | 0.5947 | 0.6705 |
| | ACC | 0.9203 | 0.9045 | 0.9024 | **0.9581** | **0.9573** | **0.9558** | 0.9149 | 0.9284 | 0.9252 | 0.9363 | 0.9257 | 0.9499 |
| | AUC | 0.8368 | 0.8045 | 0.7339 | **0.8956** | **0.8801** | **0.8856** | 0.8187 | 0.8198 | 0.8015 | 0.8516 | 0.8616 | 0.8817 |
| CR-Fill | F1 | 0.8362 | 0.7934 | 0.7595 | 0.8480 | **0.8599** | **0.8647** | 0.8372 | 0.8336 | 0.8275 | 0.7858 | 0.8036 | 0.8448 |
| | IOU | 0.6005 | 0.5213 | 0.4727 | 0.6242 | **0.6514** | **0.6571** | 0.6000 | 0.5960 | 0.5909 | 0.5104 | 0.5352 | 0.6169 |
| | ACC | 0.9382 | 0.9316 | 0.9197 | 0.9421 | **0.9535** | **0.9533** | 0.9385 | 0.9410 | 0.9445 | 0.9311 | 0.9256 | 0.9484 |
| | AUC | 0.8625 | 0.7863 | 0.7751 | 0.8639 | **0.8824** | **0.8802** | 0.8464 | 0.8399 | 0.8358 | 0.7697 | 0.8196 | 0.8429 |
| LaMa | F1 | 0.7024 | 0.5531 | 0.6255 | 0.7140 | 0.6915 | **0.7338** | 0.7236 | 0.6960 | 0.6616 | 0.6024 | 0.6500 | 0.6509 |
| | IOU | 0.3669 | 0.1312 | 0.2501 | 0.3845 | 0.3542 | **0.4309** | 0.3916 | 0.3504 | 0.3068 | 0.2132 | 0.2951 | 0.2911 |
| | ACC | 0.8992 | 0.8518 | 0.8854 | 0.8982 | 0.8944 | **0.9210** | 0.8962 | 0.9003 | 0.8975 | 0.8784 | 0.8876 | 0.8945 |
| | AUC | 0.7057 | 0.5672 | 0.6339 | 0.7119 | 0.6877 | **0.7349** | 0.7174 | 0.6891 | 0.6582 | 0.6082 | 0.6602 | 0.6515 |
| LDM | F1 | **0.7164** | 0.4827 | **0.6870** | **0.6897** | **0.6959** | **0.6681** | **0.7331** | **0.7238** | **0.6999** | 0.6248 | 0.6533 | **0.7529** |
| | IOU | **0.3936** | 0.0373 | **0.3418** | **0.3534** | **0.3613** | **0.3254** | **0.4100** | **0.4002** | **0.3628** | 0.2450 | 0.3036 | **0.4553** |
| | ACC | **0.8991** | 0.8287 | **0.8996** | **0.8875** | **0.8882** | **0.8842** | **0.8985** | **0.8961** | **0.8954** | 0.8885 | 0.8902 | **0.9200** |
| | AUC | **0.7412** | 0.5180 | **0.6975** | **0.7143** | **0.7085** | **0.6903** | **0.7483** | **0.7354** | **0.7084** | 0.6254 | 0.6670 | **0.7556** |
| SSDE | F1 | 0.6588 | 0.4806 | 0.5758 | **0.7977** | **0.8052** | **0.7910** | 0.6774 | 0.6650 | 0.6311 | 0.5658 | 0.6337 | 0.7251 |
| | IOU | 0.2974 | 0.0390 | 0.1801 | **0.5481** | **0.5608** | **0.5288** | 0.3206 | 0.3111 | 0.2613 | 0.1613 | 0.2597 | 0.4019 |
| | ACC | 0.8755 | 0.8295 | 0.8561 | **0.9337** | **0.9378** | **0.9347** | 0.8590 | 0.8700 | 0.8748 | 0.8514 | 0.8627 | 0.9049 |
| | AUC | 0.6711 | 0.5188 | 0.6004 | **0.8039** | **0.8112** | **0.7908** | 0.6984 | 0.6755 | 0.6501 | 0.5900 | 0.6525 | 0.7282 |
| DDNM | F1 | 0.6906 | 0.5389 | 0.6094 | 0.6853 | **0.7902** | **0.7664** | 0.7133 | 0.7020 | 0.6496 | 0.5446 | 0.7086 | 0.6884 |
| | IOU | 0.3494 | 0.1166 | 0.2302 | 0.3480 | **0.5310** | **0.4902** | 0.3801 | 0.3639 | 0.2914 | 0.1342 | 0.3843 | 0.3472 |
| | ACC | 0.8982 | 0.8453 | 0.8824 | 0.8955 | **0.9313** | **0.9231** | 0.8910 | 0.8961 | 0.8916 | 0.8576 | 0.8951 | 0.8977 |
| | AUC | 0.6897 | 0.5581 | 0.6198 | 0.6872 | **0.7884** | **0.7646** | 0.7118 | 0.6949 | 0.6511 | 0.5667 | 0.7200 | 0.6894 |
| RePaint | F1 | 0.6441 | 0.4870 | 0.5376 | 0.6789 | **0.7298** | **0.7295** | 0.6779 | 0.6133 | 0.6001 | 0.4828 | 0.6659 | 0.6191 |
| | IOU | 0.2750 | 0.0482 | 0.1193 | 0.3306 | **0.4171** | **0.4203** | 0.3218 | 0.2284 | 0.2143 | 0.0442 | 0.3137 | 0.2356 |
| | ACC | 0.8783 | 0.8275 | 0.8484 | 0.8859 | **0.9003** | **0.9024** | 0.8689 | 0.8561 | 0.8636 | 0.8266 | 0.8800 | 0.8694 |
| | AUC | 0.6516 | 0.5237 | 0.5600 | 0.6790 | **0.7303** | **0.7306** | 0.6864 | 0.6236 | 0.6160 | 0.5226 | 0.6799 | 0.6243 |
| Authentic | p-ACC | **0.9966** | **0.9994** | **0.9994** | 0.9878 | **0.9930** | **0.9942** | 0.9704 | **0.9924** | **0.9962** | **0.9985** | 0.9765 | **0.9948** |
| | i-ACC | **0.8000** | **0.8400** | **0.9800** | 0.7400 | 0.5200 | 0.7400 | 0.0400 | 0.5000 | **0.9600** | **0.8400** | 0.5400 | 0.7200 |

Table 4: The impact of dataset semantic correlation on performance.

| Training set | Semantic | Test set | Semantic | F1 | IOU | ACC | AUC |
|---|---|---|---|---|---|---|---|
| GRE | ✓ | CocoGlide | ✓ | 0.4570 | 0.0493 | 0.7488 | 0.5194 |
| | | COVERAGE | | 0.4748 | 0.0103 | 0.8731 | 0.4962 |
| | | IMD2020 | | 0.5038 | 0.0646 | 0.9472 | 0.5319 |
| | | Forgery ADE | | 0.6909 | 0.3406 | 0.8961 | 0.6870 |
| ADE | | CocoGlide | ✓ | 0.4785 | 0.0800 | 0.7419 | 0.5276 |
| | | COVERAGE | | 0.4971 | 0.0383 | 0.8665 | 0.5103 |
| | | IMD2020 | | 0.5704 | 0.1079 | 0.9790 | 0.5853 |
| | | Forgery ADE | | 0.7809 | 0.4975 | 0.9135 | 0.8202 |

without semantic correlation, Novozamsky et al. (2020) and the Deepfill v2 training set of Forgery ADE. Both training sets use the first 2,0000 images and employ the same pre-processing and scaling to the same size. The experimental results are shown in table 4.

It can be seen that models trained on datasets with semantic correlations and without perform similarly in dealing with semantic-related forgeries, while the model trained on datasets without semantic correlations performs significantly better in handling forged images without semantic correlations.

**Data Diversity.** This section discusses the impact of the types and number of forgery methods. We use a combination of different forgeries for training: only Deepfill v2 (Option I) or LDM (Option II), using 4 types of forgery (Option III), 6 types of forgery (Option IV), and all types of forgery (Option V).

We can see that including the corresponding samples in the training set can improve the detection performance on specific forgeries, as well as forgeries generated by similar methods. Increasing the diversity of the training forgery can significantly improve the generalization, resulting in better performance on almost all forgeries, but it can lead to an increase in the false-positive rate on authentic images.

**Negative Samples.** We further investigate the impact of introducing negative samples in training on the detection performance and false positive erros. Based on the research of VidalMata et al.

Table 5: The performance of each method on authentic images after training on datasets with and without negative samples.

| Methods | w/o Authentic Image | | w/ Authentic Image | |
|---|---|---|---|---|
| | p-ACC | i-ACC | p-ACC | i-ACC |
| ManTra-Net | 0.9334 | 0.0800 | 0.9978 | 0.6000 |
| MVSS-Net | 0.9885 | 0.6400 | 1.0000 | 1.0000 |
| PSCC-Net | 0.9363 | 0.3200 | 0.9892 | 0.9800 |
| CAT-Net | 0.9897 | 0.2200 | 0.9928 | 0.4800 |
| IID-Net | 0.9936 | 0.8000 | 1.0000 | 1.0000 |
| IF-OSN | 0.9902 | 0.5000 | 1.0000 | 1.0000 |
| IML-ViT | 0.9115 | 0.2800 | 0.9834 | 0.9800 |
| Trufor | 0.9862 | 0.6600 | 1.0000 | 1.0000 |
| FOCAL | 0.9588 | 0.0000 | 0.9602 | 0.0000 |
| EITLNet | 0.9792 | 0.6200 | 1.0000 | 1.0000 |
| GIFL | 0.9704 | 0.0400 | 0.9966 | 0.8000 |

(2023), we do not include authentic images in the training dataset (Option VI) and then set the ratio of forged images to authentic images to 1:2 (Option VII) and 2:1 (Option VIII), respectively.

Introducing a certain number of negative samples significantly reduces the occurrence of false positive errors and has no significant impact on the performance of forgery detection. Increasing the proportion of negative samples can further suppress false positive errors. However, excessive negative samples can lead to a degradation in detection performance. Therefore, we suggest adding a certain proportion of negative samples to the training samples to balance detection performance and false positive erros. Take GIFL for example, a ratio of $1:1$ between forged and authentic images is recommended.

Furthermore, we train various forgery detection methods without negative samples and with a $1:1$ ratio of negative to positive samples and evaluate their results on authentic images (Table 5). It can be observed that the false positive problem of most methods is effectively suppressed after the introduction of negative samples in training, except for FOCAL, which is limited by its contrast learning strategy that forcibly divides all images into two parts.

**Masking.** We investigate the impact of the masking post process that replaces unedited regions in forged images with authentic content. We train on the blended images and test them on both fully generated (Option IX) and blended forged images (Option XI). We also test the model trained on the complete images on the masked images (Option X).

It can be seen that the model trained on the blended image performs well on blended images but performs poorly on fully generated images, while the model trained on the complete image performs well in both cases. This is likely because the model trained on blended images will learn a shortcut that detects forgery by detecting the seam caused by blending. Therefore, we suggest training on the fully generated images to ensure that forged images, whether blended or not, can be accurately detected.

## 6 CONCLUSION

In this paper, we study the localization and detection of universal image forgeries. We propose a generalizable image forgery localization method and an efficient and robust dual-domain network. Our research emphasizes that focusing on the learning of authentic image features in pristine areas can be a generalizable way for forgery localization. Extensive experimental results indicate that our method outperforms state-of-the-art methods in locating uncounted image forgeries by a large margin and also shows competitive performance on the seen forgeries. Furthermore, we construct an image forgery dataset containing images edited by various advanced deep generative image editing methods and introduce a comprehensive training data configuration to address several data-related issues in universal image forgery detection and localization. Our training strategy and dataset configurations are independent of the model and can be applied to improve existing methods. It is worth noting that our improvement in generalization ability comes at the cost of a slight performance drop on seen forgeries, which can be an interesting problem to study in future work.

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

# A APPENDIX

## A.1 VISUALIZATION OF FEATURE ALIGNMENT

In order to visually verify the effectiveness of feature alignment, we train a fully connected layer as a decoder on the encoded features $F_I$, and decode the reconstructed features $F_r$ and target features $F_t$ into RGB images for observation, as shown in Fig. 6.

We can observe that both the reconstructed and target features have semantic and spatial structures similar to the input image after decoding, and the authentic content is reconstructed while the forged content is excluded. This confirms that the authentic features in different forged images have a high degree of consistency and are in the same feature space as the input features.

## A.2 IMPACT OF IMAGE QUALITY DEGRADATION

We apply a series of image degradation methods to the forged images, including JPEG compression with a quality factor of 25, downsampling and upsampling back with scale of 2, sharpening, mean blur with a kernel size of 7, motion blur with a kernel size of 7 and a random direction, gamma transform with a factor of 1.5, and ISO noise with a color shift factor of 0.05 and intensity factor of 0.8. We detect forged images with degraded images using the GIFL baseline, and the results are shown in Table 8. It can be seen that various image degradations are not conducive to forgery detection and may harm detection performance.

## A.3 IMPACT OF ENCODER

To investigate the influence of the pre-trained encoder's performance, we use DINOv2 ViT-L/14, DINOv2 ViT-B/14 trained on the LVD-142M dataset Oquab et al. (2023), MAE ViT-B/16 trained on ImageNet 1k He et al. (2022), and the original ViT-B/16 trained on ImageNet 21k Dosovitskiy et al. (2020) as encoders and test on the complete test set. The rest of the experimental settings and model configuration in this section are the same as those in comparative studies. The experimental results are shown in Table 7. It can be seen that using a weaker encoder will result in a decrease in the performance of the model, but it still performs significantly in detecting unseen forgeries compared to other methods in Table 1.

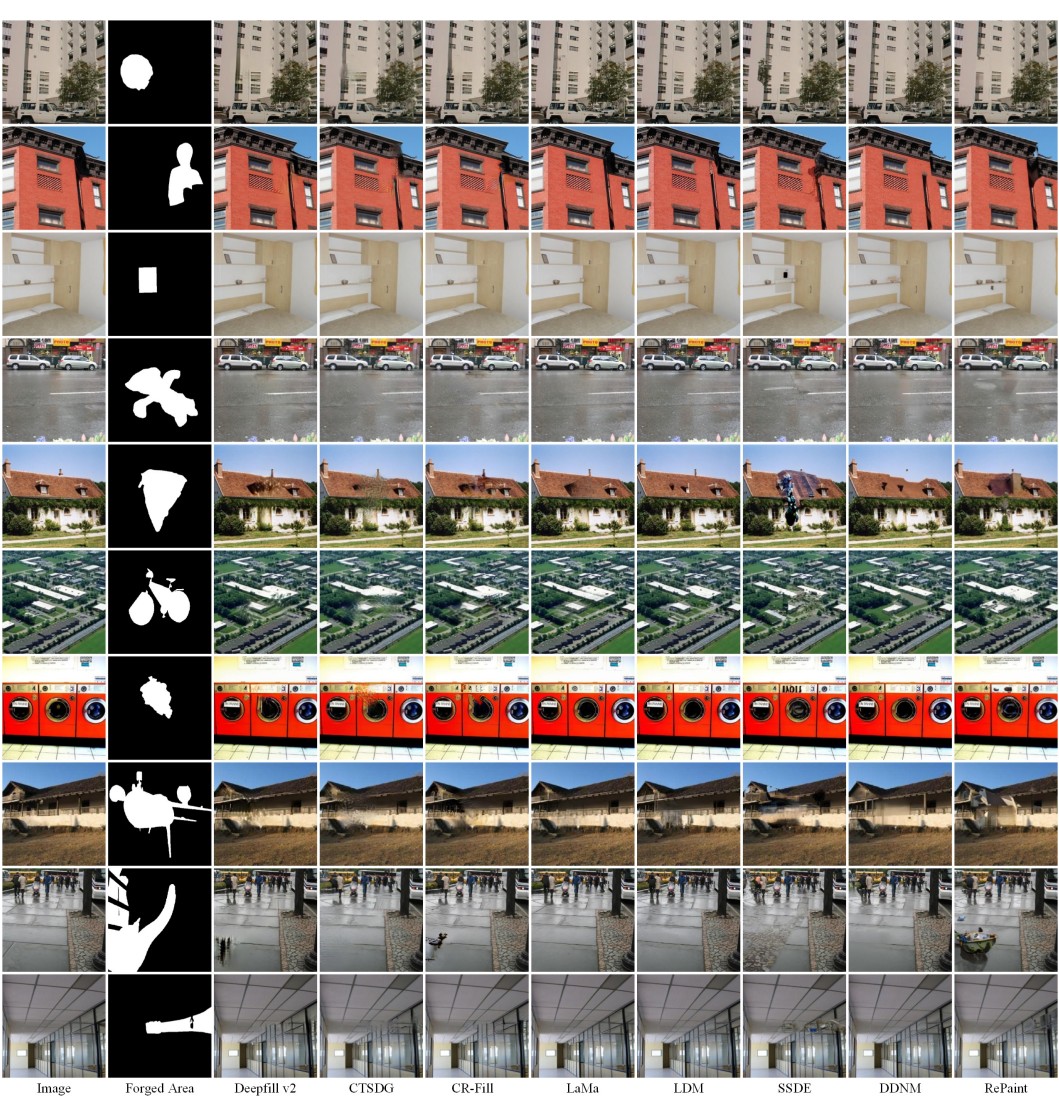

Figure 4: Forged images generated by different forgery methods in Forgery ADE dataset.

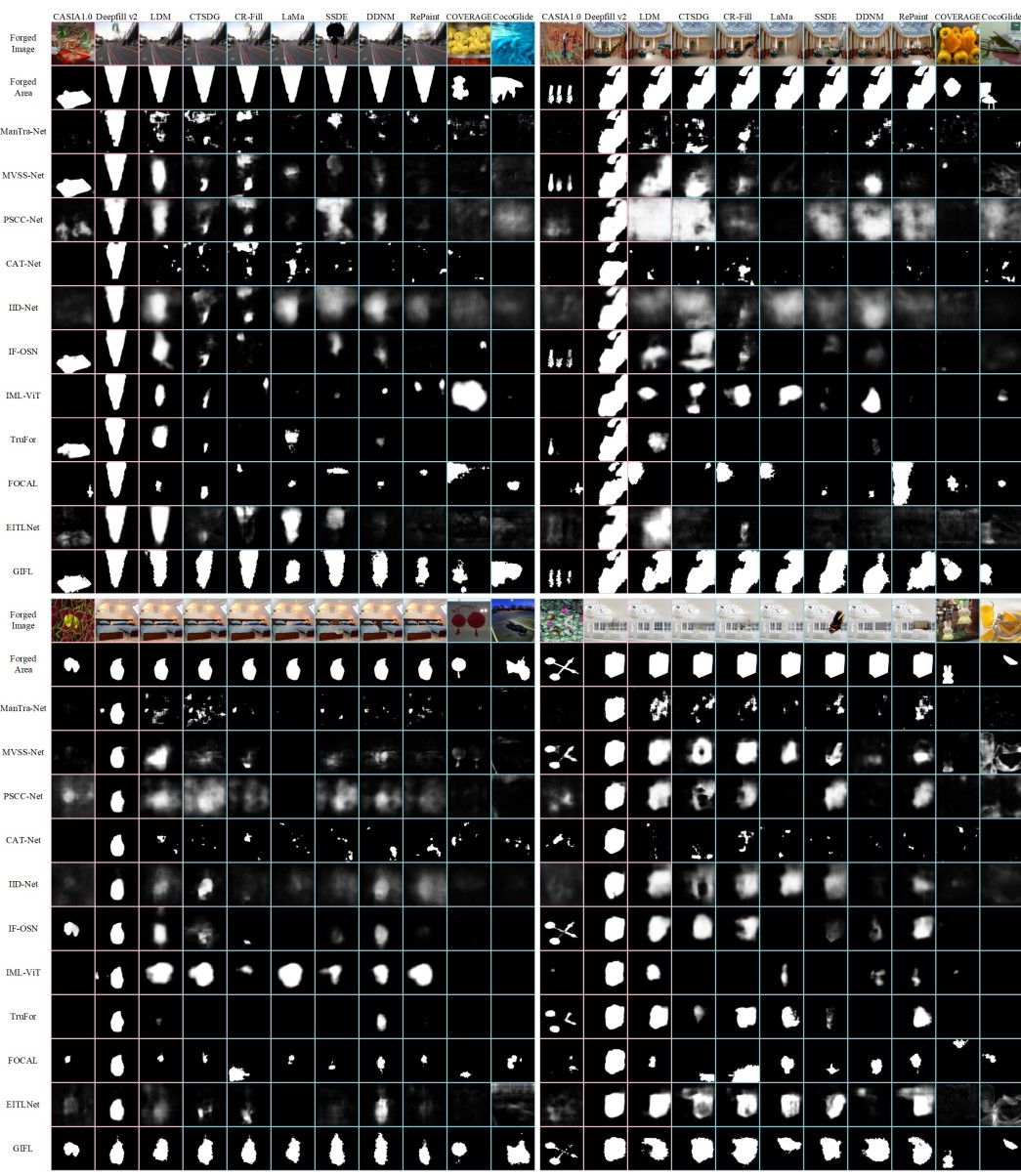

Figure 5: Visual comparison of completion images of our method and other methods. Red background for trained forgeries, blue for unseen ones. Zoom-in to see the details.

Table 6: Complete results of the ablation study.

| Prior | Forgery | Metric | Baseline | I | II | III | IV | V | VI | VII | VIII | IX |
|---|---|---|---|---|---|---|---|---|---|---|---|---|---|
| Seen | Deepfill v2 | F1 | 0.8932 | 0.9033 | 0.9007 | 0.8683 | 0.9011 | 0.8920 | 0.8421 | 0.8680 | 0.8754 | 0.8736 |
| | | IOU | 0.7195 | 0.7495 | 0.7364 | 0.6594 | 0.7341 | 0.7133 | 0.6186 | 0.6621 | 0.6759 | 0.6810 |
| | | ACC | 0.9650 | 0.9719 | 0.9681 | 0.9552 | 0.9692 | 0.9652 | 0.9289 | 0.9589 | 0.9569 | 0.9545 |
| | | AUC | 0.9093 | 0.9061 | 0.9039 | 0.8810 | 0.9054 | 0.8956 | 0.8979 | 0.8704 | 0.8947 | 0.9058 |
| | LDM | F1 | 0.7164 | 0.7391 | 0.7333 | 0.6287 | 0.7087 | 0.6823 | 0.6828 | 0.6631 | 0.6694 | 0.7100 |
| | | IOU | 0.3936 | 0.4203 | 0.4181 | 0.2535 | 0.3829 | 0.3348 | 0.3623 | 0.3018 | 0.3262 | 0.3963 |
| | | ACC | 0.8991 | 0.8973 | 0.8999 | 0.8790 | 0.8999 | 0.8951 | 0.8314 | 0.8917 | 0.8836 | 0.8635 |
| | | AUC | 0.7412 | 0.7513 | 0.7478 | 0.6432 | 0.7249 | 0.6861 | 0.7280 | 0.6707 | 0.6895 | 0.7629 |
| Unseen | CTSDG | F1 | 0.8032 | 0.7479 | 0.7819 | 0.7853 | 0.6813 | 0.7875 | 0.7904 | 0.7625 | 0.7763 | 0.7934 |
| | | IOU | 0.5464 | 0.4459 | 0.5017 | 0.5095 | 0.3318 | 0.5226 | 0.5295 | 0.4708 | 0.5017 | 0.5372 |
| | | ACC | 0.9203 | 0.9037 | 0.9120 | 0.9257 | 0.8959 | 0.9250 | 0.8969 | 0.9221 | 0.9135 | 0.8964 |
| | | AUC | 0.8368 | 0.7616 | 0.7989 | 0.8024 | 0.6723 | 0.7945 | 0.8567 | 0.7582 | 0.8095 | 0.8556 |
| | CR-Fill | F1 | 0.8362 | 0.7912 | 0.7916 | 0.8114 | 0.7097 | 0.8387 | 0.7967 | 0.7523 | 0.8011 | 0.8008 |
| | | IOU | 0.6005 | 0.5098 | 0.5102 | 0.5563 | 0.3718 | 0.6062 | 0.5346 | 0.4414 | 0.5397 | 0.5436 |
| | | ACC | 0.9382 | 0.9257 | 0.9191 | 0.9339 | 0.9103 | 0.9426 | 0.9036 | 0.9174 | 0.9285 | 0.9121 |
| | | AUC | 0.8625 | 0.7851 | 0.7858 | 0.8210 | 0.6939 | 0.8350 | 0.8523 | 0.7439 | 0.8257 | 0.8480 |
| | LaMa | F1 | 0.7024 | 0.6335 | 0.6465 | 0.6232 | 0.5327 | 0.6402 | 0.7063 | 0.5786 | 0.6874 | 0.7226 |
| | | IOU | 0.3669 | 0.2524 | 0.2679 | 0.2408 | 0.1105 | 0.2703 | 0.3797 | 0.1742 | 0.3506 | 0.4045 |
| | | ACC | 0.8992 | 0.8867 | 0.8805 | 0.8833 | 0.8573 | 0.8886 | 0.8736 | 0.8702 | 0.8991 | 0.8898 |
| | | AUC | 0.7057 | 0.6281 | 0.6399 | 0.6261 | 0.5551 | 0.6406 | 0.7260 | 0.5867 | 0.6866 | 0.7276 |
| | SSDE | F1 | 0.6588 | 0.6108 | 0.6033 | 0.6076 | 0.4916 | 0.5988 | 0.6823 | 0.5679 | 0.6691 | 0.6769 |
| | | IOU | 0.2974 | 0.2205 | 0.2138 | 0.2192 | 0.0526 | 0.2157 | 0.3608 | 0.1580 | 0.3246 | 0.3394 |
| | | ACC | 0.8755 | 0.8596 | 0.8551 | 0.8652 | 0.8340 | 0.8631 | 0.8345 | 0.8478 | 0.8837 | 0.8517 |
| | | AUC | 0.6711 | 0.6255 | 0.6226 | 0.6203 | 0.5260 | 0.6226 | 0.7350 | 0.5919 | 0.6847 | 0.7183 |
| | DDNM | F1 | 0.6906 | 0.6173 | 0.6384 | 0.5984 | 0.5225 | 0.6070 | 0.6833 | 0.5800 | 0.6641 | 0.7008 |
| | | IOU | 0.3494 | 0.2324 | 0.2605 | 0.2081 | 0.0974 | 0.2297 | 0.3646 | 0.1823 | 0.3157 | 0.3794 |
| | | ACC | 0.8982 | 0.8756 | 0.8694 | 0.8730 | 0.8472 | 0.8814 | 0.8346 | 0.8635 | 0.8924 | 0.8698 |
| | | AUC | 0.6897 | 0.6199 | 0.6451 | 0.6060 | 0.5484 | 0.6194 | 0.7281 | 0.5921 | 0.6716 | 0.7313 |
| | RePaint | F1 | 0.6441 | 0.5912 | 0.5872 | 0.5681 | 0.4708 | 0.5789 | 0.6643 | 0.5510 | 0.6199 | 0.6474 |
| | | IOU | 0.2750 | 0.1899 | 0.1919 | 0.1566 | 0.0261 | 0.1816 | 0.3223 | 0.1413 | 0.2440 | 0.2945 |
| | | ACC | 0.8783 | 0.8557 | 0.8467 | 0.8495 | 0.8284 | 0.8520 | 0.8413 | 0.8421 | 0.8583 | 0.8382 |
| | | AUC | 0.6516 | 0.6024 | 0.6079 | 0.5810 | 0.5126 | 0.5936 | 0.6967 | 0.5703 | 0.6422 | 0.6784 |

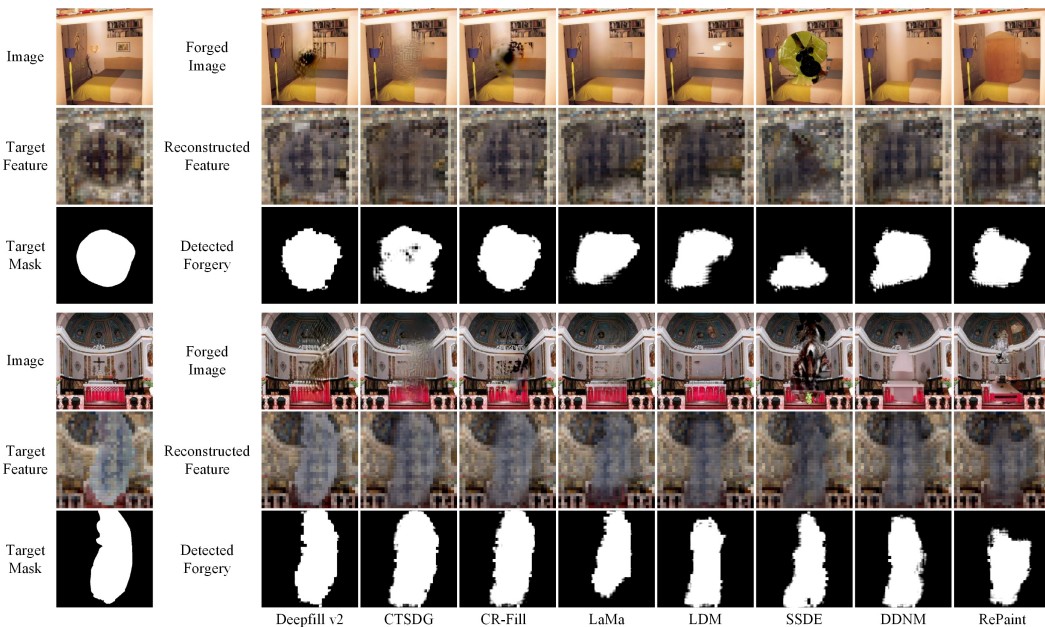

Figure 6: Visualization of target features and reconstruction features of different types of forged images. Zoom-in to see the details.

Table 7: The impact of different pre-trained encoders.

| Prior | Metric | DINOv2-L | DINOv2-B | MAE-B | ViT-B |
|-------|--------|----------|----------|-------|-------|
| Trained | F1 | 0.7815 | 0.6982 | 0.6490 | 0.6307 |
| | IOU | 0.5037 | 0.3596 | 0.3115 | 0.2486 |
| | ACC | 0.9498 | 0.9216 | 0.9332 | 0.8956 |
| | AUC | 0.7953 | 0.7258 | 0.6843 | 0.6566 |
| Unseen | F1 | 0.6703 | 0.6326 | 0.6290 | 0.6076 |
| | IOU | 0.3212 | 0.2565 | 0.2531 | 0.2165 |
| | ACC | 0.9114 | 0.8853 | 0.9008 | 0.8691 |
| | AUC | 0.6882 | 0.6664 | 0.6485 | 0.6347 |

Table 8: Detection results under various image degradation.

| Forgery | Metric | Baseline | Compression | Resizing | Sharpen | Blur | Motion Blur | Gamma | Noise |
|---------|--------|----------|-------------|----------|---------|------|-------------|-------|-------|
| Deepfill v2 | F1 | 0.8932 | 0.8552 | 0.8790 | 0.8658 | 0.8244 | 0.8821 | 0.8904 | 0.8659 |
| | IOU | 0.7195 | 0.6506 | 0.6932 | 0.6675 | 0.5937 | 0.6999 | 0.7120 | 0.6655 |
| | ACC | 0.9650 | 0.9526 | 0.9608 | 0.9544 | 0.9429 | 0.9606 | 0.9638 | 0.9569 |
| | AUC | 0.9093 | 0.8712 | 0.8977 | 0.8859 | 0.8442 | 0.8952 | 0.9064 | 0.8853 |
| CTSDG | F1 | 0.8032 | 0.7963 | 0.8041 | 0.7627 | 0.7918 | 0.8023 | 0.7889 | 0.8022 |
| | IOU | 0.5464 | 0.5336 | 0.5427 | 0.4720 | 0.5351 | 0.5415 | 0.5215 | 0.5397 |
| | ACC | 0.9203 | 0.9212 | 0.9196 | 0.9100 | 0.9268 | 0.9219 | 0.9177 | 0.9238 |
| | AUC | 0.8368 | 0.8181 | 0.8376 | 0.7798 | 0.8102 | 0.8297 | 0.8220 | 0.8218 |
| CR-Fill | F1 | 0.8362 | 0.7894 | 0.8185 | 0.7838 | 0.7486 | 0.8260 | 0.8351 | 0.7930 |
| | IOU | 0.6005 | 0.5212 | 0.5701 | 0.5100 | 0.4545 | 0.5801 | 0.5956 | 0.5253 |
| | ACC | 0.9382 | 0.9235 | 0.9315 | 0.9189 | 0.9099 | 0.9343 | 0.9409 | 0.9240 |
| | AUC | 0.8625 | 0.8175 | 0.8434 | 0.8118 | 0.7708 | 0.8457 | 0.8510 | 0.8189 |
| LaMa | F1 | 0.7024 | 0.6782 | 0.6909 | 0.6515 | 0.6583 | 0.6805 | 0.6992 | 0.6626 |
| | IOU | 0.3669 | 0.3328 | 0.3513 | 0.2908 | 0.3048 | 0.3370 | 0.3644 | 0.3066 |
| | ACC | 0.8992 | 0.8921 | 0.8944 | 0.8920 | 0.8899 | 0.8929 | 0.9019 | 0.8984 |
| | AUC | 0.7057 | 0.6879 | 0.7077 | 0.6518 | 0.6692 | 0.6900 | 0.7001 | 0.6664 |
| LDM | F1 | 0.7164 | 0.6820 | 0.7009 | 0.6988 | 0.6390 | 0.6949 | 0.7138 | 0.6869 |
| | IOU | 0.3936 | 0.3435 | 0.3696 | 0.3659 | 0.2779 | 0.3632 | 0.3885 | 0.3506 |
| | ACC | 0.8991 | 0.8861 | 0.8956 | 0.8839 | 0.8715 | 0.8940 | 0.8980 | 0.8847 |
| | AUC | 0.7412 | 0.7063 | 0.7211 | 0.7198 | 0.6648 | 0.7217 | 0.7350 | 0.7203 |
| SSDE | F1 | 0.6588 | 0.6385 | 0.6883 | 0.5895 | 0.6169 | 0.6360 | 0.6443 | 0.6695 |
| | IOU | 0.2974 | 0.2703 | 0.3464 | 0.1982 | 0.2397 | 0.2701 | 0.2726 | 0.3184 |
| | ACC | 0.8755 | 0.8670 | 0.8834 | 0.8521 | 0.8683 | 0.8708 | 0.8721 | 0.8819 |
| | AUC | 0.6711 | 0.6571 | 0.7076 | 0.6124 | 0.6322 | 0.6586 | 0.6554 | 0.6877 |
| DDNM | F1 | 0.6906 | 0.6724 | 0.7118 | 0.6656 | 0.6344 | 0.7058 | 0.6837 | 0.6619 |
| | IOU | 0.3494 | 0.3248 | 0.3810 | 0.3150 | 0.2656 | 0.3748 | 0.3372 | 0.3071 |
| | ACC | 0.8982 | 0.8949 | 0.9011 | 0.8869 | 0.8790 | 0.9017 | 0.8968 | 0.8859 |
| | AUC | 0.6897 | 0.6743 | 0.7087 | 0.6671 | 0.6473 | 0.7079 | 0.6825 | 0.6665 |
| RePaint | F1 | 0.6441 | 0.6333 | 0.6565 | 0.6123 | 0.6117 | 0.6401 | 0.6333 | 0.6248 |
| | IOU | 0.2750 | 0.2643 | 0.2961 | 0.2341 | 0.2278 | 0.2753 | 0.2572 | 0.2524 |
| | ACC | 0.8783 | 0.8711 | 0.8805 | 0.8582 | 0.8587 | 0.8710 | 0.8760 | 0.8672 |
| | AUC | 0.6516 | 0.6518 | 0.6635 | 0.6333 | 0.6287 | 0.6608 | 0.6408 | 0.6400 |
| Authentic | p-ACC | 0.9966 | 0.9904 | 0.9909 | 0.9945 | 0.9898 | 0.9909 | 0.9972 | 0.9959 |
| | i-ACC | 0.8000 | 0.6800 | 0.7200 | 0.8000 | 0.7200 | 0.6200 | 0.7200 | 0.7200 |