# OpenReview forum: "Learning Universal Features for Generalizable Image Forgery Localization"
_ICLR.cc/2025/Conference — ICLR 2025 Conference Withdrawn Submission_

### Official Review · Reviewer_EnLi · 2024-10-29

**Soundness:** 2
**Presentation:** 2
**Contribution:** 2
**Rating:** 3
**Confidence:** 5

**Summary:**

The authors propose learning universal features from the perspective of real traces to better address unknown types of forgeries. To achieve this, a new training paradigm is introduced along with the Fourier transform-based transformer UFLT. Additionally, the authors construct the ForgeryADE dataset based on eight generation algorithms. The experimental section demonstrates a certain superiority compared to existing algorithms.

**Strengths:**

- Propose a more general feature learning paradigm from a real perspective.
- A Fourier transform based transformer UFLT is established for the forgery localization task.
- Construct the ForgeryADE dataset, which contains 8 different forgery methods.

**Weaknesses:**

Methodology:
- I concur with the viewpoint that "authentic features ... are more consistent across different types of forgeries and can be used as universal features." Along this line, I think it is possible to train and extract universal authentic features using only authentic data (e.g., [1*, 2*] that use only authentic data to learn camera fingerprints to build forgery localization algorithms), which would further substantiate this view. Unfortunately, in Sec. 3.1, the authors only utilize forged data to construct the training process, lacking further discussion and analysis in this regard.

Technology:
- The UFLT proposed in Sec. 3.2 is not fully compared with existing methods (FFC, ViT, and other spectral detection networks), and lacks reasonable research motivation. For example, the statement in line 208 that "these methods can only extract the spectral traces of some specific forgeries" lacks evidence support. It is recommended that the authors add experimental phenomena to illustrate the shortcomings of existing methods and the advantages of the proposed method.

Dataset: Although the authors have created the new ForgeryADE dataset, it does not exhibit significant improvements or unique attributes for the IFL task.
- Firstly, in terms of dataset scale, it comprises only 20,000 original training images and 2,000 test images with a relatively low resolution (512x512), which falls short compared to existing large dataset like [3*].
- Secondly, in terms of the generation process, randomly selecting masks and reconstructing them using (not the latest GAN or Diffusion) generation methods is also like the existing dataset construction process.
- Lastly, regarding data quality, examples from ForgeryADE (shown in the first column of Figure 3, such as CTSDG and SSDE in the left column, and Deepfillv2 and DDNM in the right column) exhibit noticeable visual flaws. It is recommended that the authors employ a quality assessment system to filter out low-quality synthetic forgery data.
- Therefore, I do not believe ForgeryADE could stand as an independent core contribution point. Perhaps, considering creating more intriguing datasets from a multimodal perspective (such as text, speech, etc., controlling the forgery process) could be a more interesting approach.

Experiments:
- The core focus of this paper is on learning general features for unseen data. However, the training and test sets remain within the same domain ("each of these methods is applied to ... training images and test images," as mentioned in line 258). This experimental setup makes it challenging to demonstrate whether the proposed method can effectively learn sufficient cross-domain, general features.
- In the comparative experiments, the authors indicate that they retrained the models of their competitors. However, there is a lack of detailed explanation on how this retraining was conducted, which could potentially result in the competitor algorithms not achieving their intended effects. A fairer approach would involve conducting control experiments using the same training and test sets that the competitors all share. For instance, CAT-Net, FOCAL, and TruFor all utilized the same training set.

Small question:
- The original size of the images in the ForgeryADE is 512x512, so why is line 308 experimented on a size of 448x448?

[1*] “Exif as language: Learning cross-modal associations between images and camera metadata” in CVPR’23.

[2*] “Noiseprint: A CNN-based camera model fingerprint” in TIFS’19.

[3*] “Rethinking Image Editing Detection in the Era of Generative AI Revolution” in ACM MM’24.

**Questions:**

Please refer to the weaknesses.

---

### Official Review · Reviewer_SL7r · 2024-11-01

**Soundness:** 2
**Presentation:** 3
**Contribution:** 3
**Rating:** 5
**Confidence:** 4

**Summary:**

This paper introduces a forgery localization model that generalizes on unseen data distributions well called GIFL. The method focuses on learning universal features from authentic image content of a forged image rather than focusing on the part of the image that is forged, and in doing so makes the model more robust to various types of image forgeries. They initially use a frozen encoder to encode both the features of the input image (forged) and the non-forged part of the same input image. The authors then introduce a Universal Forgery Localization Transformer (UFLT) that utilizes both spatial and spectral domain information to enhance detection capabilities. The approach involves reconstructing authentic image features and aligning them with input features to improve generalization. Additionally, the researchers create a new dataset called Forgery ADE, which contains images manipulated by various advanced deep generative editing methods.

**Strengths:**

* It’s a very well written paper. The problem they try to tackle is an important problem that needs to be addressed.
* Authors introduce a novel approach to generalizable forgery detection. Trying to use representations from pristine parts of the image is a key idea.
* Model outperforms other existing state of the art models in the problem of image manipulation detection.

**Weaknesses:**

* While the model demonstrates strong generalization to unseen forgeries in the test data, outperforming other methods in this regard, it shows a slight performance drop on seen forgeries (i.e., forgeries present in the training data) compared to existing models like LDM and CTSDG. This is counterintuitive because models typically perform better on the training set (seen forgeries) than on the test set (unseen forgeries). The fact that GIFL struggles slightly more with forgeries it was trained on suggests that this may be due to the use of the same pristine images in both the training and test data. This overlap could artificially inflate performance on unseen forgeries compared to other models, as the model may learn underlying features from the pristine content rather than focusing solely on identifying manipulations. This approach might lead to improved generalization to unseen manipulations but may not be as robust in detecting forgeries in scenarios where both the image content and forgery types are more variable.
* The paper lacks clear details on how the dataset is split between training and testing, particularly regarding out-of-distribution (OOD) testing. It appears that the same images with different manipulations are used for both ID and OOD evaluations, which might blur the distinction between the two. Additionally, there is no clear indication of whether entirely unseen images (not present in the training set, even in pristine form) are included in the OOD test set. This undermines the claim of the model’s generalization ability, as the testing protocol may not fully represent a challenging OOD scenario.

**Questions:**

* In the line "To effectively exploit the traces exposed at different depths in different domains, we draw inspiration from FFC Chi et al. (2020) and establish the interaction and fusion of features across domain and across depth by connecting the paths of the two domains in each layer of the network," could you expand on what "domain" and "depth" refer to in this context? It would be helpful to clarify if "domain" refers to spatial vs. spectral representations and if "depth" is related to different layers in the transformer or network architecture.
* In Fig. 1, it appears that FtF_tFt​ is not directly an input to UFLT, but this line seems to suggest otherwise: "We then utilize a Universal Forgery Localization Transformer (UFLT) to align the features of the forged image with those of the partial image." Could you clarify how this alignment process is done beyond what’s mentioned about the loss function? Is the alignment occurring explicitly within the layers of UFLT, or is it primarily driven by the loss function guiding the final output?
* While GIFL shows notable improvements on unseen forgeries, there is a slight performance drop on seen forgery types compared to existing methods, as noted in the comparison with LDM and CTSDG. How do you explain this gap, given that GIFL performs better in many visual samples? What aspect of the network or loss function might be responsible for the lower performance on seen forgeries with these methods?
* The paper mentions the use of the same images with different manipulations for in-distribution (ID) and out-of-distribution (OOD) evaluation. Are there any images in the testing set that do not appear in the training set, even in pristine form? Specifically, do you test the model on both OOD image types and OOD manipulation types simultaneously? Clarifying how the dataset is split between training and testing would help in understanding the generalization performance.

---

### Official Review · Reviewer_5oh6 · 2024-11-03

**Soundness:** 3
**Presentation:** 2
**Contribution:** 2
**Rating:** 5
**Confidence:** 5

**Summary:**

The paper proposes a novel method for Generalizable Image Forgery Localization (GIFL) capable of detecting both known and unseen forgeries by learning universal features from unedited regions rather than relying on forgery-specific traces. The authors also introduce a new dataset, Forgery ADE, comprising images manipulated by various advanced generative models to enhance generalization in forgery detection.

**Strengths:**

1.This paper proposes a robust approach to detect unseen image forgeries by focusing on universal features rather than forgery-specific artifacts.

2.This paper introduces the Forgery ADE dataset, enhancing the applicability of the model to diverse generative editing techniques.

3.Comprehensive experimentation and performance evaluation, showcasing significant improvements in unseen forgery detection over existing methods.

4.Detailed ablation studies offer insights into the components’ contributions, highlighting the model's resilience to various forgery techniques.

**Weaknesses:**

1. The methodology of this paper closely resembles that of a Context Autoencoder (CAE), with only limited modifications and enhancements on top of the existing framework. So the approach demonstrates insufficient novelty.

2. There are already numerous studies utilizing advanced AIGC inpainting methods [1, 2] to generate large-scale image tampering datasets, making the proposed Forgery ADE dataset highly redundant with prior work. Additionally, the statement in line 243, "most existing datasets only include forged images and lack authentic images as negative samples," is inaccurate. Major IFDL datasets, such as CASIA1+ and IMD2020, do include authentic images as negative samples.

[1] TGIF: Text-Guided Inpainting Forgery Dataset.

[2] GIM: A Million-scale Benchmark for Generative Image Manipulation Detection and Localization.

3. The writing of this paper is confusing with many typos. Formulas are used irregularly. For example, M_t appearing in the formula on line 193 does not explain its meaning.  Formulas between lines lack numbering.

4. This paper does not discuss the limitations of its method or suggest future work, which makes it incomplete.

**Questions:**

Please refer to the weakness.

---

### Official Review · Reviewer_VY6w · 2024-11-04

**Soundness:** 2
**Presentation:** 2
**Contribution:** 2
**Rating:** 3
**Confidence:** 4

**Summary:**

This paper proposes a new method for image forgery localization, named UFLT. UFLT applies dual attentions to utilize information both in spatial and spectral domains for generalizable features. In addition, the paper releases an image forgery dataset based on various manipulation methods. Evaluations on the proposed dataset prove the effectiveness of UFLT.

**Strengths:**

1. The proposed method achieves promising performance on a wide variety of forgery images.

2. The authors construct a comprehensive dataset based on various representative manipulation methods of GANs and diffusion models, contributing to the community of forgery detection and localization.

3. Experiments and ablation studies are complete and detailed.

**Weaknesses:**

Weaknesses:
1.	Although the authors claim to address data-related issues in image forgery detection by introducing a training configuration, this contribution is limited and less innovative.

2.	The comparison and analysis between baselines and GIFL in Figure 1 are unfair because the advantages of existing methods are excessively omitted.

3.	The explanation in Sec. 3.2 should be revised and elaborated. Lack of critical details about the proposed. Universal Forgery Localization Transformer (UFLT): Line 214: How do paths from the spatial and spectral domains connect? Line 226: how do the vision transformer take advantage of features from the spectral domain?)

4.	In Tab. 1, in most methods, the metrics of AUC and F1 is quite poor, and not consistent with the higher ACC. I wonder if real and fake data in testing datasets are balanced, and expect an analysis and explanation about the results.

5.	Conclusions in Sec. 5.4 seems contradictory to the implementation in Sec. 5.1, raising the question for the method. Specifically, Sec.5.4 points out that removing semantic correlations benefits forgery detection. However, as detailed in Sec. 5.1, a pretrained DINOv2 is used as the encoder, which is typically trained by datasets with rich semantic information.

**Questions:**

Questions:
1.	Lacking essential explanation on the caption of Figure 2, which is the overall structure of the proposed method.

2.	In Sec. 5.1, the number and scale of the training set is not revealed.

---

### Official Review · Reviewer_G7SL · 2024-11-04

**Soundness:** 2
**Presentation:** 2
**Contribution:** 3
**Rating:** 5
**Confidence:** 5

**Summary:**

This paper aims to propose a generalizable method for detecting and localizing image forgeries, namely Generalizable Image Forgery Localization (GIFL). This method is able to detect seen and unseen image forgeries after training, providing a solution to false information in the era of adversarial generative AI. This new method focuses on learning general features from the original content, rather than traces of specific forgeries. These features are relatively consistent across different types of forgeries and can be used as general features to localize unseen forgeries.

**Strengths:**

1. The paper focuses on learning the general features of the original image instead of relying on specific forgery traces, which makes sense.
2. The author constructed a new dataset containing images edited by various deep generative models, which helps to fight against generative AI.

**Weaknesses:**

1. There is nothing uniquely novel about introducing Fourier transform and performing space-frequency interaction, which has been adopted by many previous studies in computer vision. The paper does not explain why Fourier transform is introduced and its relationship with image forgery localization. It is recommended to describe the motivation for introducing Fourier transform in detail.
2. Is the globality brought by Fourier transform meaningful for image forgery localization? The paper does not explain in detail why Fourier transform is only performed on patches.
3. Is there any overlap between the source dataset used by the unseen forged dataset and the source dataset used by the trained dataset? If so, how much of the generalizability claimed by the paper is due to the fact that the original images are the same.

**Questions:**

It is suggested that the authors give some theoretical proofs and experimental observations to demonstrate the effectiveness of the introduced Fourier transform and the designed network structure for image forgery localization.

---

### Note · Authors · 2024-11-15

I have read and agree with the venue's withdrawal policy on behalf of myself and my co-authors.